# Implications of the availability and distribution of birth weight on addressing neonatal mortality: population-based assessment from Bihar state of India

G Anil Kumar,[1] Sibin George,[1] Md Akbar,[1] Debarshi Bhattacharya,[2] Priya Nanda,[2] Lalit Dandona,[1,3] Rakhi Dandona  [1,3]

[1]Public Health Foundation of India, Gurgaon, Haryana, India
[2]Bill & Melinda Gates Foundation India, New Delhi, India
[3]Institute for Health Metrics and Evaluation, University of Washington, Seattle, Washington, USA

**Correspondence to**
Dr Rakhi Dandona;
rakhi.dandona@phfi.org

## ABSTRACT

**Objective** A large proportion of neonatal deaths in India are attributable to low birth weight (LBW). We report population-based distribution and determinants of birth weight in Bihar state, and on the perceptions about birth weight among carers.

**Design** A cross-sectional household survey in a state representative sample of 6007 live births born in 2018–2019. Mothers provided detailed interviews on sociodemographic characteristics and birth weight, and their perceptions on LBW (birth weight <2500 g). We report on birth weight availability, LBW prevalence, neonatal mortality rate (NMR) by birth weight and perceptions of mothers on LBW implications.

**Setting** Bihar state, India.

**Participants** Women with live birth between October 2018 and September 2019.

**Results** A total of 5021 (83.5%) live births participated, and 3939 (78.4%) were weighed at birth. LBW prevalence among those with available birth weight was 18.4% (95% CI 17.1 to 19.7). Majority (87.5%) of the live births born at home were not weighed at birth. LBW prevalence decreased and birth weight ≥2500 g increased significantly with increasing wealth index quartile. NMR was significantly higher in live births weighing <1500 g (11.3%; 95% CI 5.1 to 23.1) and 1500–1999 g (8.0%; 95% CI 4.6 to 13.6) than those weighing ≥2500 g (1.3%, 95% CI 0.9 to 1.7). Assuming proportional correspondence of LBW and NMR in live births with and without birth weight, the estimated LBW among those without birth weight was 35.5% (95% CI 33.0 to 38.0) and among all live births irrespective of birth weight availability was 23.0% (95% CI 21.9 to 24.2). 70% of mothers considered LBW to be a sign of sickness, 59.5% perceived it as a risk of developing other illnesses and 8.6% as having an increased probability of death.

**Conclusions** Missing birth weight is substantially compromising the planning of interventions to address LBW at the population-level. Variations of LBW by place of delivery and sociodemographic indicators, and the perceptions of carers about LBW can facilitate appropriate actions to address LBW and the associated neonatal mortality.

## STRENGTHS AND LIMITATIONS OF THIS STUDY

⇒ Data on birth weight documented for a representative sample of live births including neonatal deaths.
⇒ Documentation of birth weight based on recall, which is of reasonable quality based on the global criterion.
⇒ Perceptions of caregivers on low birth weight documented in the same population.

g) prevalence between 2012 and 2030.[1] LBW is a significant indicator of maternal and fetal health predicting mortality and stunting, and of adult-onset chronic conditions.[2–7] The global LBW prevalence was estimated at 14.6% in 2015,[8 9] and short gestation for birth weight accounted for an estimated 1.43 million deaths and 139 million disability-adjusted life years in 2017.[10]

South Asia, with India as its largest component, was estimated to have the highest LBW prevalence for any region in the world in 2015 as per the most recent global update on LBW prevalence which provided country-level estimates.[8 9] However, LBW prevalence for India was not estimated in that report due to quality concerns with the available data.[8] We have reported LBW prevalence of 21.4% in India in 2017 as part of the Global Burden of Disease (GBD) study,[11] and that 83% of neonatal deaths could be attributed to LBW in India in 2017.[12] LBW prevalence has shown modest decline over time in India, and it is projected that India is unlikely to meet the LBW national and global nutrition targets.[11] The inadequate availability and quality of birth weight data in India, like many low-income and middle-income countries, is a major hindrance in tracking LBW as a priority.[8 9 11]

In this background, we report on a population-based assessment of birth weight

## INTRODUCTION

Global nutrition targets include a 30% reduction in low birth weight (LBW, weight <2500

in the Indian state of Bihar, which is among the most populous Indian states accounting for a significant burden of neonatal mortality.[12] The LBW prevalence in Bihar was estimated as 23.4% in 2017 in the GBD study.[11] The aim of this report is to provide nuanced data for policy makers and programme planners on the availability and distribution of birth weight, and implications of birth weight non-availability on robustness of LBW estimate which is of utmost significance in planning of interventions to reduce LBW in order to address neonatal mortality. Furthermore, we present the perceptions about LBW among the carers which can improve specificity of interventions to address LBW. We use data as is without smoothening or imputation in order to highlight for the policy makers the gaps in the birth weight data that are to be addressed for meaningful action.[8 11]

## METHODS

For the survey, a state representative sample of 6000 live births was selected using a multistage sampling approach from 37 of the 38 districts of Bihar state, excluding the Lakhisarai district. In the first stage, 70 functioning community/primary health centres (CHC/PHC) were randomly sampled with probability proportional to population size from a total of 445 functioning CHC/PHCs, with each catering to an average of 84 villages. In the next stage, five villages were selected from the catchment area of each of the selected CHC/PHC using the village list available in the Census 2011.[13] To arrive at a cluster size of 300 households, villages with <300 households were combined with an adjacent village, and the large villages were split into equal-sized segments of 300 households using natural boundaries. In total, 350 clusters were sampled using a systematic sampling. Each selected cluster was mapped and all the households (a household was defined as people eating from the same kitchen) were enumerated to identify the live births delivered by women aged 15–49 years between October 2018 and September 2019.

The mother/caregiver of each identified live birth was contacted for a detailed interview irrespective of whether the baby was currently alive. Details on the sociodemography, the pregnancy, delivery and postnatal care of the eligible live birth were documented. Specifically, for the analysis reported in this paper, birth weight was recorded from the mother or caregiver of the child based on their recall. We also documented the mother/caretaker's perception of the birth weight for each live birth (very weak, weak, normal, overweight), and whether the mother/caretaker perceived low birth weight in a baby to be an indication of sickness, and if so why. Furthermore, the possible reasons for LBW in babies, how to prevent LBW and if the mother/caretaker thought if the delivery process was different based on the birth weight were also documented. The questionnaire was developed in English and then translated into Hindi (local language), after which it was back-translated into English to ensure the accurate and relevant

meaning and intent of the questions. Pilot testing of the questionnaires was carried out and modifications made as necessary. Data were collected between November 2019 and January 2020 using Open Development Kit by interviewers trained in study procedures. Data entered were scrutinised using the internal consistency checks built in to detect and correct errors using standardised procedures to meet the data quality. To further improve data quality, spot checks were conducted by the supervisors in 10% of the households and back checks were done in 15% of the households. At least three attempts were made to reach out to all the eligible live births.

We tested the quality of birth weight data by using the criteria used for the report on the global LBW prevalence estimates.[8] Poor quality data were defined as extreme heaping with >55% of all birth weights falling on three values (2500 g, 3000 g or 3500 g); >10% of births weighed at least 4500 g or excessive heaping on the tail end of the birth weight distribution with >5% of birth weights at 250–500 g and 5500 g. We report on the quality of birth weight data, and for which live births the values of 2500 g, 3000 g or 3500 g are more likely to be reported at the population-level.[8] We assessed the assumption if the data on child not weighted at birth was missing at random in this population using the Little's test for missing completely at random.[14]

We categorised birth weight into five categories for analysis: <1500 g, 1500–1999 g, 2000–2499 g, <2500 g (LBW), and 2500 g or more. We present birth weight prevalence per 100 live births for these five categories with 95% CIs, and also for not being weighted at birth, and for birth weight could not be recalled considering all live births irrespective of birth weight availability. We then report birth weight prevalence for these five birth weight categories considering only the live births for whom birth weight was available. Among these, the prevalence and mean birth weight with SD is also reported by maternal age, maternal education, wealth index, sex of the baby, length of the pregnancy, place of delivery and based on live birth survival. Wealth index was estimated using the standard questions and methods used in the National Family Health Survey.[15] Multiple logistic regression was run to investigate the association of LBW among the live births with birth weight available with the above variables with all the variables introduced simultaneously in the model. ORs with 95% CIs are presented for the regression analysis.

We explored the association of neonatal and postneonatal mortality with birth weight. Based on the difference in neonatal mortality rates between live births for whom birth weight was available versus those for whom birth weight was not available, we also report proportionately adjusted LBW prevalence in those with birth weight available to estimate the LBW prevalence in those with birth weight not available. In addition, a variety of perceptions of the caregivers about LBW are reported. All analysis was performed using STATA V.13.1 software (StataCorp, USA).

## Patient and public involvement

Patients or the public were not involved in the design, or conduct, or reporting, or dissemination plans of our research.

## RESULTS

We identified 6007 live births representative of the Bihar state between October 2018 and September 2019 from 5852 women aged 15–49 years in 55 475 households. Detailed interview was available for 5021 (83.6%) live births, majority (98.2%) of whom were singleton births, 2614 (52.1%) were boys, 2870 (57.2 %) were born in a public health facility and 150 (3%) were currently not alive. Of the 5021 live births, 3939 (78.4%) were weighed at birth but birth weight could not be recalled for 292 (7.4%, 95% CI 6.6 to 8.3) live births.

## Quality of birthweight data

Considering the 3647 live births with birth weight available, 52% of all birth weight values fell on 2500 g, 3000 g or 3500 g; 1.6% live births weighed at least 4500 g and 0.36% of birth weights were either at 250–500 g or 5500 g. This indicates data to be of reasonable quality, as the heaping was less than the criteria for poor quality data. Significant variation was seen in the reporting of birth weight values of 2500 g, 3000 g and 3500 g by maternal age ($\chi^2$, p=0.008), maternal education ($\chi^2$, p<0.001) and place of delivery ($\chi^2$, p=0.028) as shown in online supplemental figure 1. The data on child not weighted at birth were not missing completely at random (p<0.001).

## Distribution of birth weight among all live births

Considering all live births irrespective of birth weight availability, prevalence of birth weight ≥2500 g was 59.3% (95% CI 57.9 to 60.6), <2500 g of LBW was 13.3% (95% CI 12.4 to 14.3) and of live births not weighed at birth was 21.5% (95% CI 20.4 to 22.7) as shown in online supplemental table 1. Importantly, the prevalence of live births not weighed at birth was 87.5 (95% CI 85.4 to 89.3) in home births as compared with only negligible facility births for whom birth weight was not measured (online supplemental table 1).

## Distribution of birth weight among live births with birth weight available

Among live births with birth weight available, the mean birth weight was 2848.2 g with SD of ±647.2 g (table 1), and was significantly lower for live births born at 6–7 months of gestation (1710.6±577.4 g) and for live births of younger mothers aged <20 years (2718.0±642.5 g). Girls, live births belonging to lower wealth index quartile and live births who did not survive were significantly more likely to have a lower mean birth weight as compared with boys, those belonging to higher wealth index quartile and those currently alive, respectively (table 1).

**Table 1** Mean birth weight for live births between October 2018 and September 2019 for whom birth weight could be recalled in the Indian state of Bihar

| | Total | Availability of birth weight (% of total) | Mean birth weight (g) |
|---|---|---|---|
| Overall | 5021 | 3647 (72.6) | 2848.2±647.2 |
| Maternal age (years)*† | | | |
| 15–19 | 529 | 407 (76.9) | 2718.0±642.5 |
| 20–24 | 2392 | 1808 (75.6) | 2836.6±646.3 |
| 25–29 | 1453 | 1028 (70.8) | 2911.8±632.8 |
| ≥30 | 633 | 392 (61.9) | 2878.7±662.5 |
| Maternal education†§ | | | |
| No education | 1907 | 1172 (61.5) | 2801.0±685.6 |
| Classes 1–5 | 760 | 544 (71.6) | 2826.0±664.4 |
| More than class 5 | 2350 | 1928 (82.0) | 2885.4±613.3 |
| Wealth index quartile†¶ | | | |
| I | 1255 | 777 (61.9) | 2781.9±690.1 |
| II | 1255 | 861 (68.6) | 2800.7±656.0 |
| III | 1255 | 945 (75.3) | 2879.9±659.2 |
| IV | 1255 | 1063 (84.7) | 2907.0±588.0 |
| Sex‡ | | | |
| Boy | 2614 | 1939 (74.2) | 2888.7±647.1 |
| Girl | 2407 | 1708 (71.0) | 2802.3±644.3 |
| Gestation period (months)† | | | |
| 6–7 | 46 | 33 (71.7) | 1710.6±577.4 |
| 8 | 944 | 701 (74.3) | 2735.7±631.7 |
| >8 | 4027 | 2910 (72.3) | 2889.7±635.2 |
| Birth order† | | | |
| First | 1366 | 1110 (81.3) | 2775.2±628.5 |
| Second | 1369 | 1019 (74.4) | 2892.5±653.1 |
| More than second | 2282 | 1515 (66.4) | 2874.8±649.8 |
| Place of delivery†§ | | | |
| Public sector facility | 2870 | 2622 (91.4) | 2839.3±625.9 |
| Private sector facility | 1022 | 890 (87.1) | 2880.7±697.0 |
| Home | 1125 | 132 (11.7) | 2839.2±679.6 |
| Current status of live birth‡ | | | |
| Died on day 0 of birth | 57 | 26 (45.6) | 2644.2±1082.1 |
| Died between day 1 and 27 of birth | 58 | 40 (69.0) | 2611.3±1071.3 |
| Died between day 28 and 11 months of age | 35 | 22 (62.9) | 2368.2±771.9 |
| Alive | 4871 | 3559 (73.1) | 2855.3±634.4 |

*Data not available for 14 live births.
†$\chi^2$ test of significance, p<0.001.
‡$\chi^2$ test of significance, p=0.001.
§Data not available for 4 live births.
¶Data not available for 1 live birth.

The prevalence of LBW was 18.4 (95% CI 17.1 to 19.7), and that of birth weight <1500 g was 1.5 (95% CI 1.1 to 1.9), of 1500–1999 g was 4.1 (95% CI 3.5 to 4.8) and of 2000–2400 g was 12.8 (95% CI 11.8 to 13.9) as shown in table 2. LBW prevalence was 5.6 times higher among

**Table 2** Prevalence of birth weight by categories among the live births who had birth weight available for select characteristics in the Indian state of Bihar for live births between October 2018 and September 2019

| | Prevalence per 100 live births (95% CI) | | | | |
| --- | --- | --- | --- | --- | --- |
| | Birth weight ≥2500 g | Birth weight <2500 g | Birth weight 2000–2499 g | Birth weight 1500–1999 g | Birth weight <1500 g |
| Overall | 81.6 (80.3 to 82.9) | 18.4 (17.1 to 19.7) | 12.8 (11.8 to 13.9) | 4.1 (3.5 to 4.8) | 1.5 (1.1 to 1.9) |
| Maternal age (years)* | | | | | |
| 15–19 | 73.0 (68.4 to 77.1) | 27.0 (22.9 to 31.6) | 19.9 (16.3 to 24.1) | 6.1 (4.2 to 8.9) | 1.0 (0.4 to 2.6) |
| 20–24 | 81.5 (79.6 to 83.2) | 18.5 (16.8 to 20.4) | 13.0 (11.5 to 14.6) | 4.0 (3.2 to 5.1) | 1.5 (1.0 to 2.2) |
| 25–29 | 86.0 (83.7 to 88.0) | 14.0 (12.0 to 16.3) | 9.2 (7.6 to 11.2) | 3.3 (2.4 to 4.6) | 1.5 (0.9 to 2.4) |
| ≥30 | 80.6 (76.4 to 84.2) | 19.4 (15.8 to 23.6) | 13.8 (10.7 to 17.6) | 4.3 (2.7 to 6.9) | 1.3 (0.5 to 3.0) |
| Maternal education† | | | | | |
| No education | 78.7 (76.2 to 80.9) | 21.3 (19.1 to 23.8) | 14.0 (12.1 to 16.1) | 5.1 (4.0 to 6.5) | 2.2 (1.5 to 3.2) |
| Class 1–5 | 79.2 (75.6 to 82.4) | 20.8 (17.6 to 24.4) | 14.5 (11.8 to 17.7) | 5.2 (3.6 to 7.4) | 1.1 (0.5 to 2.4) |
| More than class 5 | 84.2 (82.5 to 85.8) | 15.8 (14.2 to 17.5) | 11.6 (10.3 to 13.1) | 3.2 (2.5 to 4.1) | 1.0 (0.6 to 1.5) |
| Wealth index quartile‡ | | | | | |
| I | 78.7 (76.2 to 80.9) | 23.0 (20.2 to 26.1) | 15.3 (13.0 to 18.0) | 5.8 (4.4 to 7.7) | 1.9 (1.2 to 3.2) |
| II | 79.2 (75.6 to 82.4) | 20.8 (18.2 to 23.6) | 14.1 (11.9 to 16.5) | 4.9 (3.6 to 6.5) | 1.9 (1.1 to 3.0) |
| III | 84.2 (82.5 to 85.8) | 17.7 (15.4 to 20.2) | 13.2 (11.2 to 15.5) | 3.2 (2.2 to 4.5) | 1.3 (0.7 to 2.2) |
| IV | 78.7 (76.2 to 80.9) | 13.6 (11.7 to 15.8) | 9.6 (8.0 to 11.5) | 3.1 (2.2 to 4.3) | 0.9 (0.5 to 1.7) |
| Sex | | | | | |
| Boy | 84.0 (82.3 to 85.5) | 16.0 (14.5 to 17.7) | 10.8 (9.5 to 12.2) | 4.0 (3.2 to 4.9) | 1.3 (0.9 to 1.9) |
| Girl | 79.0 (77.0 to 80.9) | 21.0 (19.2 to 23.0) | 15.1 (13.5 to 16.9) | 4.3 (3.4 to 5.3) | 1.6 (1.1 to 2.4) |
| Gestation period (months) | | | | | |
| 6–7 | 12.1 (4.6 to 28.5) | 87.9 (71.5 to 95.5) | 24.2 (12.5 to 41.8) | 36.4 (21.8 to 54.0) | 27.3 (14.7 to 45.0) |
| 8 | 74.6 (71.3 to 77.7) | 25.4 (22.3 to 28.8) | 17.7 (15.0 to 20.7) | 5.9 (4.3 to 7.9) | 1.9 (1.1 to 3.2) |
| >8 | 84.2 (82.8 to 85.5) | 15.8 (14.5 to 17.2) | 11.5 (10.4 to 12.7) | 3.3 (2.7 to 4.0) | 1.0 (0.7 to 1.4) |
| Birth order | | | | | |
| First | 78.1 (75.6 to 80.5) | 21.9 (19.6 to 24.4) | 15.4 (13.4 to 17.7) | 5.2 (4.1 to 6.7) | 1.3 (0.8 to 2.1) |
| Second | 83.4 (81.0 to 85.6) | 16.6 (14.4 to 19.0) | 12.0 (10.1 to 14.1) | 3.2 (2.3 to 4.5) | 1.4 (0.8 to 2.3) |
| More than second | 83.2 (81.2 to 85.0) | 16.8 (15.0 to 18.8) | 11.5 (10.0 to 13.2) | 3.8 (3.0 to 4.9) | 1.5 (1.0 to 2.3) |
| Place of delivery | | | | | |
| Public sector facility | 81.9 (80.4 to 83.3) | 18.1 (16.7 to 19.6) | 13.2 (12.0 to 14.6) | 3.6 (2.9 to 4.4) | 1.3 (0.9 to 1.8) |
| Private sector facility | 81.7 (79.0 to 84.1) | 18.3 (15.9 to 21.0) | 11.4 (9.4 to 13.6) | 5.4 (4.1 to 7.1) | 1.6 (0.9 to 2.6) |
| Home | 78.0 (70.1 to 84.3) | 22.0 (15.7 to 29.9) | 14.4 (9.4 to 21.5) | 5.3 (2.5 to 10.7) | 2.3 (0.7 to 6.8) |

**Table 2** Continued

| | Prevalence per 100 live births (95% CI) | | | | |
| --- | --- | --- | --- | --- | --- |
| | Birth weight ≥2500 g | Birth weight <2500 g | Birth weight 2000–2499 g | Birth weight 1500–1999 g | Birth weight <1500 g |
| Current status of live birth | | | | | |
| Died on day 0 of birth | 61.5 (41.7 to 78.2) | 38.5 (21.8 to 58.3) | 11.5 (3.7 to 30.8) | 19.2 (8.1 to 39.2) | 7.7 (1.9 to 26.6) |
| Died between day 1 and 27 of birth | 55.0 (39.4 to 69.7) | 45.0 (30.3 to 60.6) | 17.5 (8.5 to 32.6) | 17.5 (8.5 to 32.6) | 10.0 (3.8 to 24.0) |
| Died between day 28 and 11 months of age | 59.1 (37.7 to 77.5) | 40.9 (22.5 to 62.3) | 13.6 (4.3 to 35.5) | 18.2 (6.8 to 40.3) | 9.1 (2.2 to 30.7) |
| Alive | 82.2 (80.9 to 83.4) | 17.8 (16.6 to 19.1) | 12.8 (11.7 to 13.9) | 3.8 (3.2 to 4.4) | 1.3 (1.0 to 1.7) |

*Data not available for 12 live births.
†Data not available for 3 live births.
‡Data not available for 1 live birth.

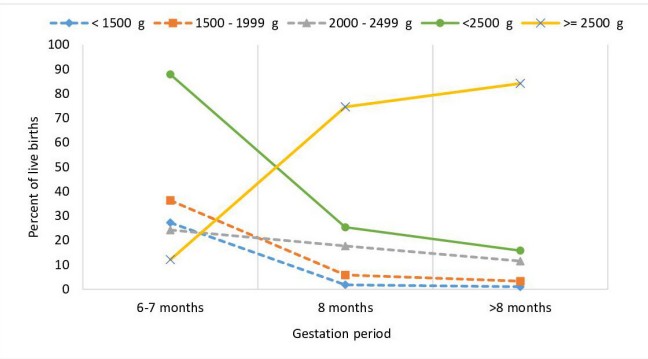

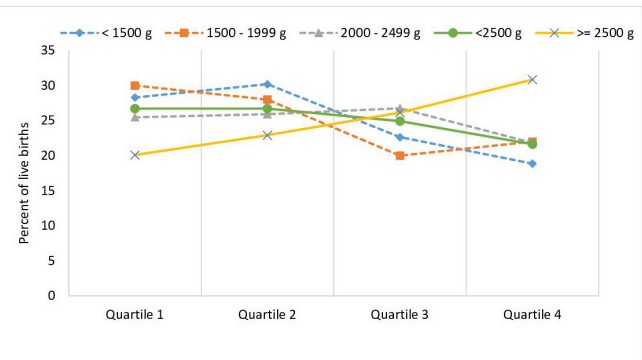

**Figure 1** Distribution of birth weight by the gestation period and wealth index quartile for live births between October 2018 and September 2019 for whom birth weight was available in the Indian state of Bihar.

the babies who were born with 6–7 months of gestation as compared with those born >8 months of gestation (table 2 and figure 1). LBW prevalence decreased and that for birth weight ≥2500 g increased significantly (p<0.001) with increasing wealth index quartile (table 2 and figure 1). Using multiple logistic regression (online supplemental table 2), the most significant odds of having LBW were for live births with gestation period of 6–7 months (OR 34.0; 95% CI 11.6 to 99.6).

Of the 670 LBW babies, the parents of 463 (69.1%) live births were informed by the health provider that the baby was weak/LBW. This proportion was 87.2% for the 203 live births with birth weight of <2000 g and 94.3% for 53 live births with birth weight of <1500 g. Considering the 190 facility live births with birth weight <2000 g, live births at public facility (84%) were significantly less likely to be informed by the health provider of the baby being weak/having LBW as compared with those born in a private sector facility (93.6%; Z test for significance p<0.1).

### Mortality and birth weight

A total of 150 (3.0%) live births were not currently alive of whom 114 (76%) had died during the neonatal period (table 1). The neonatal mortality rate in live births weighing <1500 g (11.3%; 95% CI 5.1 to 23.1) and 1500–1999 g (8.0%; 95% CI 4.6 to 13.6) was significantly higher than in those weighing ≥2500 g (table 3). The neonatal mortality rate in live births for whom birth weight was

**Table 3** Mortality by birth weight categories among the live births born between October 2018 and September 2019 in the Indian state of Bihar

| Birth weight | Number of live births | Number of neonatal deaths | Neonatal mortality rate, % (95% CI) | Number of deaths in postneonatal period to 11 months of age | Postneonatal mortality rate to 11 months of age, % (95% CI) |
|---|---|---|---|---|---|
| ≥2500 g | 2977 | 38 | 1.3 (0.9 to 1.7) | 13 | 0.4 (0.3 to 0.8) |
| <2500 g | 670 | 28 | 4.2 (2.9 to 6.0) | 9 | 1.3 (0.7 to 2.6) |
| <1500 g | 53 | 6 | 11.3 (5.1 to 23.1) | 2 | 3.8 (0.9 to 14.0) |
| 1500–1999 g | 150 | 12 | 8.0 (4.6 to 13.6) | 4 | 2.7 (1.0 to 6.9) |
| 2000–2499 g | 467 | 10 | 2.1 (1.2 to 3.9) | 3 | 0.6 (0.2 to 2.0) |
| Birth weight available | 3647 | 66 | 1.8 (1.4 to 2.3) | 22 | 0.6 (0.4 to 0.9) |
| Not recalled | 292 | 15 | 5.1 (3.1 to 8.4) | 0 | 0 |
| Not weighed at birth | 1082 | 33 | 3.0 (2.2 to 4.3) | 14 | 1.3 (0.8 to 2.2) |
| Birth weight not available | 1374 | 48 | 3.5 (2.6 to 4.6) | 14 | 1.0 (0.6 to 1.7) |
| All live births | 5021 | 114 | 2.3 (1.9 to 2.7) | 96 | 0.7 (0.5 to 1.0) |

not available (3.5; 95% CI 2.6 to 4.6) was almost twice as high as compared with those for whom birth weight was available (1.8%, 95% CI 1.4 to 2.3) as shown in table 3. Based on this 93% higher neonatal mortality rate among live births for whom birth weight was not available, and assuming a direct correspondence between neonatal mortality rate and LBW, we estimated that LBW among live births for whom birth weight was not available would be 35.5% (95% CI 33.0 to 38.0), that is, 93% higher than the 18.4% LBW among live births for whom birth weight was available. Based on the proportions of these two groups among all live births, we estimated an overall LBW of 23.0% (95% CI 21.9 to 24.2) among all live births.

### Respondent's perceptions about LBW

Mothers were the predominant respondent in the survey (99.8%). Figure 2 shows the perception of mothers on the birth weight of their live birth. Overall, 74.7% (3748) of all mothers of live birth, 88.1% (2,622) of mothers of live births ≥2500 g and 25.5% (170) of mothers of LBW live births perceived their newborns to be of normal weight. Perception of weak or very weak was higher in LBW live births (73.3%) as compared with ≥2500 g live births (11%). Among the 53 live births with birth weight <1500 g, 36 (67.9%) were perceived to be very weak, 9 (17%) weak and 6 (1.3%) of normal weight by the mother. These perceptions are not mutually exclusive.

A total of 3527 (70.2%) mothers considered LBW to be a sign of sickness/illness. Among these 3527 women, 2988 (84.2%) perceived it as a risk of developing other illnesses, 1764 (50%) considered it a risk for weak growth and 433 (12.3%) perceived it as having an increased probability of death (not mutually exclusive). Among the 1350 (26.9%) women who did not consider LBW to be a sickness in a newborn, 1308 (96.9%) felt that the

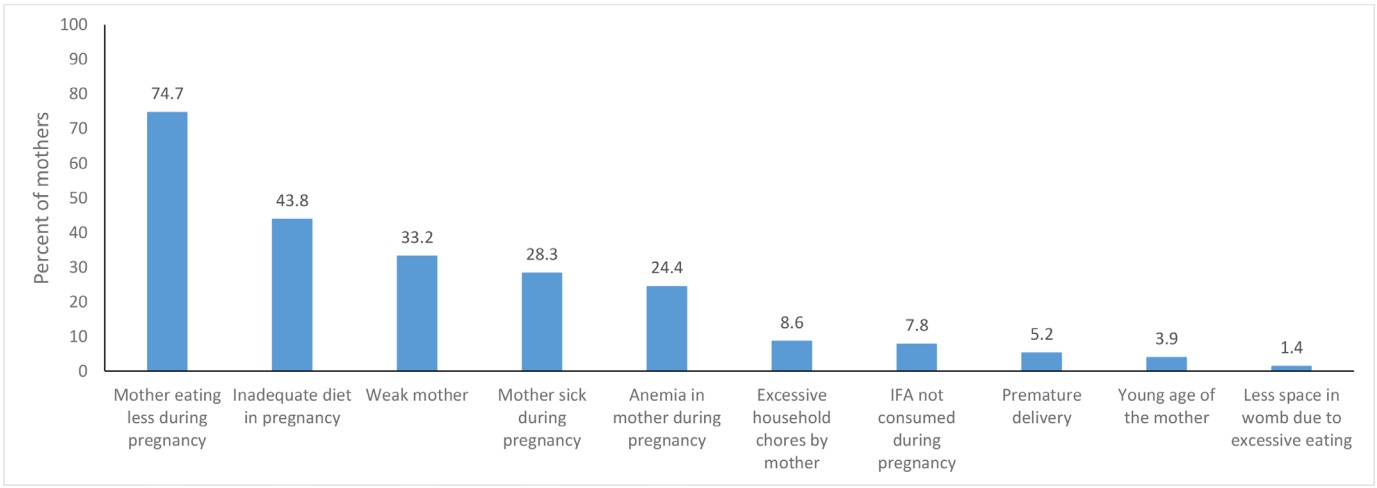

**Figure 2** Factors perceived as responsible for low birth weight in babies among the mothers of live births between October 2018 and September 2019 in the Indian state of Bihar. IFA, iron and folic acid.

baby would gain weight after birth and hence there was nothing to worry. Majority (4570; 91%) of the mothers thought that LBW baby needed extra care; and the extra care practices commonly reported (not mutually exclusive) were oil massage (76.4%), exclusive breast feeding (74.3%), seeking healthcare advice (46.6%) and keeping the baby warm (31.2%).

Figure 2 shows the possible reasons of LBW as reported by the mothers (not mutually exclusive). Mother eating less during pregnancy (74.7%), inadequate diet during pregnancy (43.8%) and weak mother (33.2%) were the most cited reasons for LBW baby. Majority of the mothers (94.9%) reported that intake of nutritious diet during prepregnancy and during pregnancy can prevent LBW, followed by full antenatal care check-up (28.3%) and iron and folic acid intake (23.3%). A total of 3026 (60.8%) mothers perceived the delivery process to be different depending on the birth weight of baby; 2515 (83.1%) felt that delivery of LBW baby was easier than that of a normal weight baby, 891 (29.4%) thought that caesarean section was needed less for LBW babies and 874 (28.9 %) felt that duration of labour was shorter for them (not mutually exclusive).

## DISCUSSION

We present the estimates for birth weight prevalence across various categories in the Indian state of Bihar, including LBW prevalence which is essential for tracking progress towards the national and global nutrition targets. These estimates are presented in two ways—including and excluding live births based on birth weight availability—to highlight the need for improved birth weight availability to arrive at robust understanding of LBW prevalence for appropriate action both within the health system and the community. Sociodemographic distribution of live births for whom birth weight was not available can facilitate formulating specific actions in these populations to improve birth weight availability. Notably, the perceptions of mothers regarding reasons for LBW and its implications can provide a framework for developing relevant actions to improve care of LBW babies and possible actions to reduce LBW prevalence.

Birth weight was missing for one out of four live births in this population. Extrapolating our findings to the estimated 2.5 million live births in 2019 in Bihar, 543 000 live births were not weighted at birth and recall was not available for 146 600. Although home births accounted for only 22% of all live births in this population, these accounted for majority of the live births who were not weighted at birth. Therefore, until facility births can be increased further in the long-term that could result in increased birth weight measurement, tracking LBW as a priority target is not possible unless urgent targeted efforts are made in the short-term to engage with the health providers who assist with home births to improve birth weight availability.

Overall, birth weight data in our study was of reasonable quality as per the criteria used in the recent report on global estimation of LBW prevalence.[8] Unlike other reports,[8 9] we did not smoothen the data for heaping, but have presented data as is to enhance understanding of where heaping was more likely to be reported to facilitate development of targeted approach in addressing this heaping. For the policy makers and programme planners, it is imperative to note where most action is needed to improve robustness of birth weight estimates. One of the assumptions made in the recent global report on LBW prevalence was that missing birth weights are missing at random and that the true distribution of birth weights in a population can be approximated by a mixture of two normal distributions.[8] Our data have highlighted that birth weight is not missing at random but in specific subgroups, and this may be need to be taken into account in assumptions for global estimates.

The LBW prevalence estimated was 18.4% considering only live births with birth weight available, and 23% in all live births by proportionately adjusting for those who did not have birth weight available based on their higher neonatal mortality rate. Even though the adjustment made for neonatal mortality is fairly simplistic, the extent of variation in LBW prevalence with this adjustment conveys the enormous implications of non-availability of birth weight for the planning of interventions and to appropriately allocate resources to address LBW at the population-level. Those without birth weight accounted for one-third of all neonatal deaths, and birth weight availability was less than half for the live births who had died on day 0. Importantly, the LBW prevalence was estimated to be almost twice among live births for whom birth weight was not available versus those for whom birth weight was available. This finding is of significance as we have previously reported that 50% of all neonatal mortality in the state to be in 0–2 days of birth, with 35% of them not weighted at birth.[16] Although the current study included only live births, our previous work in Bihar has also documented birth weight non-availability at 85% for stillbirths.[17] One of the proposed newborn quality of care indicator at health-facility level in low-income and middle-income setting is facility neonatal mortality rate disaggregated by birth weight.[18] With majority of births now in the facilities, urgent and sustained effort is needed to track this quality indicator on a routine basis, which is currently not tracked in the Indian health information system. Interestingly, the Civil Registration System captures the birth weight for all births but those data are not available in public domain to comment on availability and quality of that data.[19] As LBW and short gestation are the predominant risk factors for neonatal mortality in India and in Bihar,[12] ensuring birth weight is measured for all live births irrespective of survival at birth is extremely important. Understanding the health providers' perspectives on the need of birth weight measurement and quality is an understudied issue,[20]

and effort to improve this understanding is needed urgently to improve birth weight documentation.

A significant focus of neonatal health programmes is on caring for the small and sick newborns, and communication with the carer/family is an integral part for their meaningful participation.[21] Seven in 10 carers of LBW babies were informed by the health provider that the baby was weak/LBW, and this proportion increased with decreasing birth weight. Some additional effort is needed in the public sector facilities as the families of babies born there were less likely to be informed than those in the private sector and informing birth weight and its implications by them to the family. Importantly, 70% of the mothers interviewed considered LBW to be a sign of sickness/illness, and such level of awareness could be translated into demand for availability of birth weight in the community, and to increase uptake of relevant interventions for LBW babies.[22-27]

The finding of decrease in prevalence of LBW and increase in birth weight ≥2500 g with increasing wealth index quartile is not surprising, given that maternal undernutrition is associated with poor maternal-fetal outcomes including LBW.[2-6 28] Despite decades of efforts in India to tackle malnutrition, it was the predominant risk factor for under-5 deaths in every state of India in 2017, accounting for 68.2% of the total under-5 deaths.[11] Globally, India has the highest prevalence of body mass index <16 in women, with less prevalence in women belonging to higher wealth index.[29] Evidence from Bangladesh suggests that low levels of women's empowerment are associated with maternal undernutrition as well as with delivering LBW babies, and empowerment is lower in women of lower wealth index.[28] What is noteworthy is that majority of the women in our study were well aware of the link between maternal nutrition and LBW, highlighting that facilitators are needed to translate this awareness into action to improve maternal nutrition, which can be achieved by bringing convergence of variety of nutrition-related activities of various government ministries and stakeholders for maternal health across the life cycle.[11 30-34]

Documentation of birth weight based on recall in this study could be considered a limitation; however, these data were of reasonable quality using the global criterion.[8] The strengths of our study include an attempt to estimate LBW for all live births at the population level, and inclusion of carer perspectives in addition to birth weight availability that can facilitate actionable interventions or further implementation research to improve tracking of LBW, which is a priority global health indicator.

## CONCLUSION

Significant efforts are needed in India beyond what has been done so far to increase the availability and quality of birth weight in order to improve robustness of LBW estimates, which can help planning of appropriate interventions and investments to address this important risk factor of neonatal mortality. Without robust birth weight estimates, India may not be able to address neonatal mortality effectively to meet the Sustainable Development Goal by 2030.

**Contributors** RD and GAK had full access to data in the study, take full responsibility for the integrity of data and accuracy of the data analysis and had final responsibility for the decision to submit for publication. RD, GAK and LD conceptualised the study. RD guided the data analysis and drafted the manuscript. SG performed data analysis. MA guided data collection. MA, DB, PN and LD contributed to data analysis and interpretation. RD is responsible for the overall content as the guarantor. All authors approved the final manuscript.

**Funding** This work was supported by Bill & Melinda Gates Foundation (grant number INV-007989).

**Competing interests** PN and DB are employees of Bill & Melinda Gates Foundation. Other authors declare no completing interests.

**Patient and public involvement** Patients and/or the public were not involved in the design, or conduct, or reporting, or dissemination plans of this research.

**Patient consent for publication** Not applicable.

**Ethics approval** The ethics approval for this study was provided by the Institutional Ethics Committee of Public Health Foundation of India (study number TRC-IEC 418/19). Written informed consent was obtained from all respondents who could read and write, and the information sheet and consent form were explained by the interviewer to those who could not read/write and their thumb impressions were obtained.

**Provenance and peer review** Not commissioned; externally peer reviewed.

**Data availability statement** Data are available on reasonable request.

**ORCID iD**
Rakhi Dandona http://orcid.org/0000-0003-0926-788X

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
