## [Reviewer comments · BMJ Open]

ARTICLE DETAILS

TITLE (PROVISIONAL)	Implications of the availability and distribution of birthweight on addressing neonatal mortality: Population-based assessment from Bihar state of India
AUTHORS	KUMAR, ANIL; George, Sibin; Akbar, Md.; Bhattacharya, Debarshi; Nanda, Priya; Dandona, Lalit; Dandona, Rakhi

VERSION 1 – REVIEW

REVIEWER	Laopaiboon, M Faculty of Public Health, Khon Kaen University, Thailand, Biostatistics and Demography
REVIEW RETURNED	14-Mar-2022

GENERAL COMMENTS	The paper aim was unclear. There is not any information in the method could be understood how the authors estimated not available birthweights. The results are difficult to follow, and some parts do not answer the study aim. Most of discussion are not supported by the study results. I find difficult to follow this paper and could not give comment in details.
---

REVIEWER	Kamal, A Lahore College for Women University, Statistics
REVIEW RETURNED	16-Apr-2022

GENERAL COMMENTS	This study addressed important child health issue particularly in developing countries where NMR is high and birthweight can be one of potential factor affecting it. Overall quality of write up is satisfactory. Need to revise for some minor grammatical errors. Manuscript may be accepted after incorporation of suggestions given below. 1. Odd ratio for place of delivery (home/on route) is very high (532). So I suggest to revisit analysis and find out its reason and revise that analysis accordingly. Generally this happen sometimes due to small sample , missing observation or due to few number in particular category (132 in your case is small as compared to frequency of other categories of that variable) etc. There is not any theoretical justification for such a big odd ratio so rectify this part of analysis to handle problem. 2. On page 15 in discussion section it is mentioned that "our data has highlight that birthweight is not missing at random". How concluded without showing and results related to it after statistical analysis. Have you tested assumption of randomness? if yes , also report results. I have not seen results in manuscript. 3. Conclusion can be separated from discussion. Policy implication is missing.
---

VERSION 1 – AUTHOR RESPONSE

Reviewer: 1

The paper aim was unclear. There is not any information in the method could be understood how the authors estimated not available birthweights. The results are difficult to follow, and some parts do not answer the study aim. Most of discussion are not supported by the study results. I find difficult to follow this paper and could not give comment in details.

- The aim of the paper is mentioned on page 4, paragraph 3.
- The birthweights were documented based on recall of the mother or primary care giver (mentioned on page 5, paragraph 3, lines 3-7). When the respondent informed that birthweight was not measured, it was considered as child not weighted at birth; when the respondent could not recall birthweight, it was recorded as birthweight not recalled.
- The results are in line with the analysis mentioned in the methods, pages 6-7.
- Discussion broadly deals with improving availability of birthweight measurement in home deliveries, and implications of the study findings on understanding LBW prevalence and its association with neonatal mortality. These are the themes that are also presented in the results section.

Reviewer: 2

This study addressed important child health issue particularly in developing countries where NMR is high and birthweight can be one of potential factor affecting it. Overall quality of write up is satisfactory. Need to revise for some minor grammatical errors. Manuscript may be accepted after incorporation of suggestions given below.

1. Odd ratio for place of delivery (home/on route) is very high (532). So I suggest to revisit analysis and find out its reason and revise that analysis accordingly. Generally, this happen sometimes due to small sample, missing observation or due to few number in particular category (132 in your case is small as compared to frequency of other categories of that variable) etc. There is not any theoretical justification for such a big odd ratio so rectify this part of analysis to handle problem.

We have checked the data and have arrived at the same results. Typically, such unusually high odds ratio result from small sample as indicated by the reviewer. The issue here is that one of the categories, home births, has a very high proportion of livebirths who were not weighed at birth (87.5%) as compared with minimal proportions in the facility births as shown below (previous supplementary table 2).

	Total N=5,021 (% of total)	% of livebirths not weighed at birth
Public sector facility	2870 (57.2)	38 (1.3)
Private sector facility	1022 (20.4)	59 (5.8)

Home/on route	1125 (22.4)	984 (87.5)
---------------	-------------	------------

Given this concern, we reviewed in detail the other associations in this particular logistic regression and those are quite insignificant given the distribution of these livebirths for home deliveries. This high prevalence of livebirths not weighted at birth in home deliveries is already indicated in Supplementary Table 1 (21.5% of all livebirths compared with 87.5% for home livebirths). Therefore, these logistic regression results are not needed and we have now deleted these from the revised manuscript.

1. On page 15 in discussion section it is mentioned that "our data has highlight that birthweight is not missing at random". How concluded without showing and results related to it after statistical analysis. Have you tested assumption of randomness? if yes, also report results. I have not seen results in manuscript.

Thank you for highlighting this oversight. We have now added the details in methods (page 6, paragraph 2) and results (page 8, paragraph 2).

1. Conclusion can be separated from discussion.

This has now been separated.

1. Policy implication is missing.

We do not have a separate paragraph for policy implications as these are indicated with the respective themes that are discussed, such as:

- Page 1, paragraph 2 – “Therefore, until facility births can be increased further in the long-term that could result in increased birthweight measurement, tracking LBW as a priority target is not possible unless urgent targeted efforts are made in the short-term to engage with the health providers who assist with home births to improve birthweight availability.”
- Page 15, paragraph 2 – “One of the proposed newborn quality of care indicator at health-facility level in low- and middle-income setting is facility neonatal mortality rate disaggregated by birth weight. With majority of births now in the facilities, urgent and sustained effort is needed to track this quality indicator on a routine basis, which is currently not tracked in the Indian health information system.”
- Page 15, paragraph 2 – “Understanding the health provider’s perspectives on the need of birthweight measurement and quality is an understudied issue, and effort to improve this understanding is needed urgently to improve birthweight documentation.”
- Page 15, paragraph 2 – “What is noteworthy is that majority of the women in our study were well aware of the link between maternal nutrition and LBW, highlighting that facilitators are needed to translate this awareness into action to improve maternal nutrition, which can be achieved by bringing convergence of variety of nutrition-related activities of various government ministries and stakeholders for maternal health across the life cycle.”

Thank you for considering our manuscript. If more information were needed, I would be pleased to respond.

VERSION 2 – REVIEW

REVIEWER	Kamal, A Lahore College for Women University, Statistics
REVIEW RETURNED	20-May-2022

GENERAL COMMENTS	Run test is used to test randomness. Kindly search methods to assess assumption of missing completely at random. Little's test for missing completely at random may be used. Kindly provide exact page number, line number and table number in the letter to reviewer to easy locate corrections. I am unable to locate paragraphs or lines pointed in the context of policy implications. Also mention table or paragraphs pointed by reviewer with page number and line and corrections done with correct page number, paragraph number and line number. After these minor corrections, paper can be accepted.
---

VERSION 2 – AUTHOR RESPONSE

25 May 2022

Dear Editor,

Thank you for inviting revision of our manuscript. We have revised the manuscript by addressing the Reviewer comments as detailed below.

Reviewer: 2

1. Run test is used to test randomness. Kindly search methods to assess assumption of missing completely at random. Little's test for missing completely at random may be used.

Thank you for this suggestion from which we have also learnt. We have now used the Little's test for missing completely at random. This is now updated on page 6 (paragraph 2, lines 7-8) and page 8 (paragraph 2, lines 7-8).

1. Kindly provide exact page number, line number and table number in the letter to reviewer to easy locate corrections.

We had indicated page and paragraph number in our previous response and stated that these were in reference to the manuscript with track changes. We had not provided line number, apologies.

1. I am unable to locate paragraphs or lines pointed in the context of policy implications. Also mention table or paragraphs pointed by reviewer with page number and line and corrections done with correct page number, paragraph number and line number.

We had provided the following response in our previous letter (shown in bold). We have now updated the previous response with the page and line number as in the latest revised manuscript.

We do not have a separate paragraph for policy implications as these are indicated with the respective themes that are discussed, such as:

- Page 14, paragraph 2, lines 5-8 – “Therefore, until facility births can be increased further in the long-term that could result in increased birthweight measurement, tracking LBW as a priority target is not possible unless urgent targeted efforts are made in the short-term to engage with the health providers who assist with home births to improve birthweight availability.”
- Page 15, paragraph 2, lines 13-16 – “One of the proposed newborn quality of care indicator at health-facility level in low- and middle-income setting is facility neonatal mortality rate disaggregated by birth weight. With majority of births now in the facilities, urgent and sustained effort is needed to track this quality indicator on a routine basis, which is currently not tracked in the Indian health information system.”
- Page 15, paragraph 2, lines 20 continuing on to page 16 – “Understanding the health provider’s perspectives on the need of birthweight measurement and quality is an understudied issue, and effort to improve this understanding is needed urgently to improve birthweight documentation.”
- Page 16, paragraph 2, lines 9-13 – “What is noteworthy is that majority of the women in our study were well aware of the link between maternal nutrition and LBW, highlighting that facilitators are needed to translate this awareness into action to improve maternal nutrition, which can be achieved by bringing convergence of variety of nutrition-related activities of various government ministries and stakeholders for maternal health across the life cycle.”

1. After these minor corrections, paper can be accepted.

Thank you.

Thank you for considering our manuscript. If more information were needed, I would be pleased to respond.

Best regards,

Rakhi